# Preliminary Investigation into the Prevalence of G6PD Deficiency in a Pediatric African American Population Using a Near-Patient Diagnostic Platform

**DOI:** 10.3390/diagnostics13243647

**Published:** 2023-12-12

**Authors:** Van Leung-Pineda, Elizabeth P. Weinzierl, Beverly B. Rogers

**Affiliations:** Department of Pathology and Laboratory Medicine, Children’s Healthcare of Atlanta, and Emory University School of Medicine, Atlanta, GA 30322, USA; van.pinedawung@choa.org (V.L.-P.); elizabeth.weinzierl2@choa.org (E.P.W.)

**Keywords:** G6PD deficiency, digital microfluidics, near-patient testing, pediatric, moderate complexity, rapid

## Abstract

Glucose-6-phosphate dehydrogenase (G6PD) deficiency is prevalent in the African American population. We identified eighteen G6PD-deficient samples (9%) in a study of residual, de-identified whole blood specimens from 200 African American pediatric patients using a point-of-care instrument. This highlights the possibility of a rapid time to result for G6PD testing, which can be valuable in some clinical scenarios.

## 1. Introduction

Glucose-6-phosphate dehydrogenase (G6PD) is a housekeeping enzyme involved in the generation of NADPH and is an essential coenzyme for the protection of mature red blood cells against oxidative damage. G6PD deficiency is an X-linked disorder and the most common enzyme deficiency in humans, affecting over 400 million people worldwide, with a high prevalence in persons of African, Asian, and Mediterranean descent. Ethnicity alone does not always determine G6PD deficiency risk, as G6PD deficiency has been found in individuals outside of these higher risk populations. For instance, one study found at least 35 males with severe G6PD deficiency from one European family [1]. Females with intermediate G6PD levels can also be impacted.

G6PD deficiency is polymorphic, with more than 300 known genotypic variants, and presents as a spectrum of disease severity. G6PD-deficient individuals are susceptible to acute hemolytic anemia, with certain triggers such as ingesting food containing vicine or covicine (such as fava beans, peas, and lentils) or when exposed to certain infections (ex. viral or bacterial infections, pneumonia, or hepatitis) or drugs (methylthioninium chloride, primaquine, rasburicase, etc.) [2,3].

In G6PD-deficient adults, acute hemolysis due to triggers should be carefully monitored, as the hemolysis can be severe enough to warrant blood transfusion. Infants and children with G6PD deficiency are more vulnerable to serious adverse effects; kidney failure, severe brain damage, and even death have been reported for G6PD-deficient children after exposure to known triggers [4,5,6,7]. For newborns, G6PD deficiency increases the risk for hyperbilirubinemia and can lead to kernicterus and permanent brain damage. G6PD-deficient newborns are less likely to respond well to conventional treatment methods for hyperbilirubinemia and are at greater risk for permanent brain damage or death during infancy [8]. 

G6PD deficiency results in a lifelong susceptibility to acute hemolytic anemia and chronic hemolytic anemia, which can lead to frequent hospitalizations, organ damage, and a reduced overall quality of life in severely affected individuals. Hemolytic anemia is also triggered by certain antibiotics and medications that are used to treat malaria, including chloroquine and primaquine [9]. Patients being treated for tumor lysis syndrome with rasburicase are also at risk for developing hemolytic anemia and methemoglobinemia [10]. Because of these factors, it is important that G6PD activity be measured in at-risk patients. According to WHO guidelines [9], males and females that have less than 30% of the adjusted male median (AMM) normal enzymatic activity are considered G6PD-deficient, males that have enzymatic activity greater than 30% of the AMM are normal, females that have enzymatic activity between 30% to 80% of the AMM are considered to be intermediate, and females with enzymatic activity greater than 80% of the AMM are considered to be normal.

The early identification of G6PD deficiency and patient education regarding safe and unsafe medications and foods are vital to prevent serious complications later in life. The American Academy of Pediatrics recommends testing for G6PD enzyme activity in newborns with severe hyperbilirubinemia who are to receive phototherapy and whose family history, ethnicity, or geographic origin are known risks for G6PD deficiency, and those who are not responding well to phototherapy [11,12]. New York State has passed legislation (NY State Senate Bill S4316) for pre-discharge screening for G6PD deficiency in newborns meeting certain criteria in order to minimize adverse effects from severe hyperbilirubinemia during the newborn period and to provide education for lifelong disease management [13,14]. The U.S. Food and Drug Administration (FDA) recommends testing for G6PD prior to treatment of tumor lysis syndrome with rasburicase, but the absence of a rapid G6PD test makes this recommendation difficult to follow [10].

With this report, we present the findings of a study to detect G6PD deficiency in an African American cohort of pediatric patients at Children’s Healthcare of Atlanta. The objective of this study was to establish the prevalence of G6PD deficiency among African American newborns and pediatric patients receiving care at Children’s Healthcare of Atlanta using a moderate-complexity test (as determined by the US Food and Drug Administration) on a device with a 17 min time to result. Only a few studies have determined the expected prevalence of G6PD deficiency in African Americans. The U.S. Army performed a retrospective study of 63,302 soldiers and found prevalence rates of 12.2% and 4.1% for African American males and females, respectively [15]. Kaplan et al. (2004) found a similar prevalence of 12.8% among 500 newborn African American males [16]. Additionally, we describe the potential application of this technology when a rapid result is needed. With the American Academy of Pediatrics’s guideline on testing any newborn with recalcitrant bilirubin levels for G6PD deficiency, and the difficulty of current testing methods to identify G6PD deficiency prior to rasburicase therapy, an accessible testing platform would benefit hospitals looking to implement a test with a rapid result [10,17]. 

## 2. Materials and Methods

Residual, de-identified whole blood specimens from pediatric African American patients, their race based on information from the medical record, were used in a retrospective study to test the level of G6PD activity. This study was approved by the institutional review board (IRB) at Children’s Healthcare of Atlanta, Egleston, and was determined to meet the criteria for a waiver of informed consent. Samples were selected randomly from available leftover residual whole blood specimens. Prior to use in the study, samples were stripped of any patient identifiers and given a non-traceable research sample ID. Race, date of collection, age, and gender information were collected from the residual blood samples.

### 2.1. Sample Collection 

The samples (*n* = 200) were collected over an 11-month timespan from September 2021 to August 2022. Inclusion criteria for the samples included the following: residual pediatric specimens (<21 years old) from African American patients and collected less than 48 h before testing for G6PD deficiency. Samples were collected in either lithium heparin- or EDTA-coated tubes. The age of the subject at sample collection ranged from 10 days old to 20 years old. 

### 2.2. G6PD Enzymatic Activity Measurement

To test for G6PD, a quantitative method based on a digital microfluidic platform was used. This method required only 50 μL of sample for analysis. The digital microfluidic platform, FINDER^®^, consists of an instrument that measures 10.6 by 8.9 by 12.3 cubic inches with an 8-inch detachable commercial tablet, and disposable, single-use assay cartridges (Baebies, Inc., Durham, NC, USA). The instrument contains a spectrophotometer-based absorbance detector for colorimetric assays and a single-channel fluorescence detector. The instrument and cartridge for enzyme testing of neonatal and pediatric samples was previously described by Sista et al. [18]. A prototype instrument was previously used by Dr. Vinod Bhutani’s laboratory at Stanford University to explore neonatal screening of G6PD deficiency [19,20]. Although at the time of this study the FINDER was not yet FDA-cleared, the digital microfluidic platform for G6PD testing for neonatal hyperbilirubinemia subsequently received 510(k) clearance from the FDA [21]. 

For this study, the FINDER instrument was placed in the central core laboratory. De-identified, residual lithium heparin or EDTA specimens were mixed to homogenize the sample, and 50 µL of sample were loaded directly onto the cartridge via a sample port designed for capillary blood samples. The cartridge contains dried spots of reagents; liquid buffers are sealed in reservoirs on the cartridge and are used to reconstitute dried reagents during the automated enzyme assay protocol. A barcode with the lot number, expiration date, and calibration details was also included on the cartridge. After the cartridge containing the sample was inserted into the instrument, the assay protocol was initiated from the tablet interface and the entire assay panel, including reagent and sample preparation, subsequently ran autonomously without any additional user intervention. The whole blood was incubated with β-nicotinamide adenine dinucleotide phosphate (NADP) and glucose-6-phosphate; G6PD enzymatic activity was determined by measuring the rate of NADPH production via fluorescence and normalized to hemoglobin absorbance. 

## 3. Results

A total of 200 de-identified whole blood specimens were collected; 103 specimens were from female African American patients and 97 were from male African American patients (Table 1). The overall median age of all the patients was 11 years, with a range in age from 10 days old to 20 years old. The female specimens represented a median age of 12 years old, with a range in age from 17 days old to 20 years old. The male specimens represented a median age of 11 years old, with a range from 10 days old to 19 years old.

Table 2 highlights the G6PD activity results from the study. The adjusted male median (AMM) activity was determined to be 10.8 U/gHb by calculating the median of all the activity values from the males with values greater than 6 U/gHb (considered to be normal). As mentioned before, the males and females with an enzymatic activity less than 30% of the AMM were considered to be deficient and the males with enzymatic activity greater than 30% of the AMM were considered to be normal. Similarly, females with an enzymatic activity between 30% and 80% of the AMM were considered to be intermediate and those with an enzymatic activity greater than 80% of the AMM were considered to be normal. Thirty percent of the AMM was calculated to be 3.2 U/gHb and 80% of the AMM to be 8.6 U/gHb. The median G6PD value for all the specimens was 10.5 ± 4.0 U/gHb, the median G6PD value for the female specimens was 10.7 ± 3.5 U/gHb, and the median G6PD value for the male specimens was 10.3 ± 4.4 U/gHb.

Of the 200 samples, 15 males and 3 females were identified as G6PD-deficient by the FINDER G6PD assay. In addition, we identified 20 females (12.6%) as having intermediate G6PD activity (Figure 1). Of the deficient samples, the median G6PD value for all the specimens was 1.4 ± 0.5 U/gHb, the median G6PD value for the female specimens was 1.3 ± 0.4 U/gHb, and the median G6PD value for the male specimens was 1.5 ± 0.6 U/gHb. The histogram depictions (Figure 1) of the G6PD enzyme activity by gender follow a normal distribution, with a majority of the normal and intermediate cases falling between 6 and 16 U/gHb. 

We determined a prevalence of G6PD deficiency of 9% for the overall cohort of the African American newborn and pediatric specimens, with a prevalence of 2.9% in the females and a higher prevalence of 15.5% in the males. These results are comparable to those observed by Chinevere et al. (2006) [15], with an overall prevalence rate of G6PD deficiency in African American adults of 10.2%, with 4.1% in females and 12.2% in males. 

## 4. Discussion

The near-patient FINDER digital microfluidic G6PD assay was used to estimate the prevalence of G6PD deficiency in a pediatric African American cohort at Children’s Healthcare of Atlanta. A near-patient testing device is defined as any device that is not intended for self-testing but is intended to perform testing outside a clinical laboratory environment generally near to the patient by a healthcare professional. Prior studies reporting G6PD analysis using the Finder focused on analytical factors such as accuracy and reproducibility [19,20]. Our study further expanded on the information about this assay by performing the assay in a hospital clinical laboratory, prospectively acquiring samples, and determining the prevalence of G6PD deficiency in our population.

There is potential for a device such as FINDER to assist in the implementation of the new American Academy of Pediatrics’s guidelines for G6PD testing in newborns with recurring high bilirubin concentrations [17], as well as testing for G6PD deficiency prior to rasburicase therapy [10]. Based on this study, we found that the prevalence of G6PD deficiency in an African American pediatric cohort was comparable to previous studies [15]. Our results reinforce that there may be a significant population of African American patients, especially male, that may not be known to be G6PD-deficient, and that this condition is not being considered when medical decisions are being made about their care [10,22]. 

Our study found a similar proportion of individuals with intermediate G6PD activity in males and females (~12%). A test that provides quantitative enzymatic activity values, such as the one used in this study, could also help identify females who can serve as carriers for the disease, or may be affected clinically through their intermediate status. The limitations of this study include the limited number of samples, the self-reporting of race, and that samples came from individuals that were receiving care in the hospital, which may not reflect the general population.

Future studies are planned to correlate whether G6PD deficiency in this population is associated with abnormal liver function tests and to establish the true G6PD prevalence in the African American community in the United States. The availability of an FDA-cleared platform for rapid G6PD testing with low sample volumes could be critical to the screening success of newborns and pediatric patients for G6PD deficiency. We propose to expand this study to include a population-wide study, explore links to other health outcomes, increased readmissions, anemia, and other sequelae, and incorporate testing for all patients.

## Figures and Tables

**Figure 1 diagnostics-13-03647-f001:**
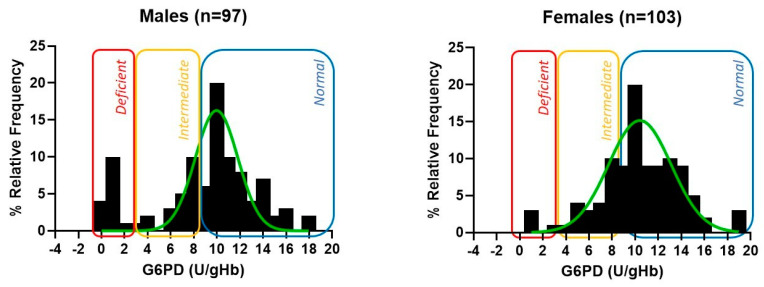
Activity distribution of males and females. Histogram depictions for G6PD enzyme activity by gender follow a normal distribution, with a majority of the normal and intermediate cases falling between 6 and 16 U/gHb.

**Table 1 diagnostics-13-03647-t001:** Patient demographics from the retrospective analysis of 200 de-identified whole blood specimens from African American pediatric patients at Children’s Healthcare of Atlanta. Abbreviations include y (years) and d (days).

Patient Demographics	
**All (*n* = 200)**	
Median Age	11 y
Low Age	10 d
High Age	20 y
Female (*n* = 103)	
Median Age	12 y
Low Age	17 d
High Age	20 y
Male (*n* = 97)	
Median Age	11 y
Low Age	10 d
High Age	19 y

**Table 2 diagnostics-13-03647-t002:** G6PD activity results from the retrospective analysis of 200 de-identified whole blood specimens from African American pediatric patients at Children’s Healthcare of Atlanta.

G6PD Activity			
**All Samples (*n* = 200)**			
	**All**	**Female**	**Male**
Median G6PD value (U/gHb)	10.5	10.7	10.3
Mean G6PD value (U/gHb)	10.2	10.9	9.4
SD of G6PD values (U/gHb)	4.0	3.5	4.4
Deficient sample (*n*)	18	3	15
Deficiency frequency	9.00%	2.91%	15.46%
**Samples with Normal and Intermediate Results**			
Median G6PD value (U/gHb)	10.8	10.8	10.8
Mean G6PD value (U/gHb)	11.0	11.2	10.9
SD of G6PD values (U/gHb)	3.0	3.2	2.9
**Samples with Deficient Results**			
Median G6PD value (U/gHb)	1.4	1.3	1.5
Mean G6PD value (U/gHb)	1.4	1.5	1.4
SD of G6PD values (U/gHb)	0.5	0.4	0.6

## Data Availability

The data that support the findings of this study are available on request from the corresponding author (BBR). The data are not publicly available because, although de-identified, they are derived from patient samples.

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
