# Peer review of "Preliminary Investigation into the Prevalence of G6PD Deficiency in a Pediatric African American Population Using a Near-Patient Diagnostic Platform"

_diagnostics, 2023, doi:10.3390/diagnostics13243647_

Round 1

Reviewer 1 Report

Comments and Suggestions for Authors

The authors showcase the outcomes of a study where they used the FINDER digital microfluidic platform to detect G6PD (Glucose 6-phosphate dehydrogenase) deficiency in a cohort of African-American Pediatric patients. They show that of the 200 samples (103 female, 97 male) tested, 15 males and 3 females were identified as G6PD deficient. The results presented were comparable to a prior retrospective study where the prevalence of G6PD deficiency was 12.2% among adult African-American males and 4.2% among adult African-American females.

This study is among the initial efforts to evaluate G6PD levels in pediatric African-American patients using the FINDER platform. However, the uniqueness of the research is somewhat diminished because the platform was already employed for neonatal G6PD screening in prior research. To truly showcase the system's value, the authors are encouraged to conduct further studies that correlate the platform's measurements with health outcomes, as recommended in the ‘Discussion’ section of the manuscript.

 Minor concerns: 

  1. Line 104 has a typo when referring to the volume of blood required for analysis
  2. Figure 1 is too small in its current form and is not readable
  3. Typo in Table 2 under ‘Normal and Intermediate’ SD of G6PD values

Author Response

Thank you for the careful review. We have corrected the typographical error in line 104 and I can't find the typographical error in Table 2. Possibly the copy editor will see it. As far as the figure being small, there is nothing we can do about that, but hope that the reader will increase the magnification of the manuscript to see it, as the publication is online. We do intend to do additional studies with this instrument to determine outcomes, etc. It is quite a good platform, and we have the patient population.

Reviewer 2 Report

Comments and Suggestions for Authors

The definition of "Near Patient" testing should be added, something like: "A NPT device is defined as any device that is not intended for self-testing but is intended to perform testing outside a laboratory environment, generally near to, or at the side of, the patient by a health care professional."

The total number of specimens should be mentioned in the abstract, absolute number of positives makes no sense without this. The introduction is long and detailed, it gives a good description of this deficiency. It would be interesting to know how the authors moved from digging into files to ascertain ethnicity to their goal to de-identify the subjects of this study.

Author Response

Thank you for the detailed review. We added a sentence of further define near patient testing in line 185 and we thank the reviewer for the verbiage, which we used most of. We mentioned the number of total samples in the abstract. We didn't go into the details of de-identification, but the IRB was written to review the chart, place a code on the sample and data from the chart, and then de-link the original data from the test code.

Reviewer 3 Report

Comments and Suggestions for Authors

Glucose-6-phosphate dehydrogenase deficiency is one of the most common enzymopathies. Its prevalence is about 10% in some populations, e.g., people of African or Mediterranean origin. The authors gave a good introduction about the background of the medical significance of the deficiency. In some clinical cases, it is necessary the fast and simple testing of glucose-6-phosphate dehydrogenase deficiency. The report gives an example for the use of a recently developed device applying for fast detection of this deficiency on 200 pediatric blood samples obtained from African American patients. The results fit well to the known data

There are some minor mistakes that should be corrected in the proof:

Methods

line 104 50 L - -50 μL

References

Ref 1

Blood Cells Mol Dis (2021);92. - Blood Cells Mol Dis (2021);92:102625.

Ref 6

East Mediterr Heal J (2017);23. - East Mediterr Heal J (2017);23:28-30. doi: 10.26719/2017.23.1.28.

Ref 9

Guide to G6PD deficiency rapid diagnostic testing to support P. vivax radical cure - Guide to G6PD deficiency rapid diagnostic testing to support P. vivax radical cure (P. vivax italicus)

Ref 19

Semin Perinatol (2020):151356. - Semin Perinatol (2021) 45:151356.

Author Response

Thank you We have made all of the corrections indicated.

Reviewer 4 Report

Comments and Suggestions for Authors

My major question is what's new in your work? The device and its application had been already described (Wong et al., 2020; Bhutani et al., 2015). The prevalence of G6PD deficiency had been already described as well. So what's the value of your work? It's seems just to be a proof of the fact that in your hospital the device as also working and the prevalence of G6PD deficiency is the same as elsewhere.

Comments on the Quality of English Language

The English needs minor revisions.

Author Response

We thank the reviewer for the comments. It is true that what we did was to use this device on a patient population in a clinical laboratory, obtained expected results, and provided a framework to do further studies. The previous article by Wong was using a prototype, and was a study to identify instrument and test characteristics. We are now implementing it in the clinical laboratory. We believe that is enough of a difference. 

We prefer not to have English editing.

Round 2

Reviewer 1 Report

Comments and Suggestions for Authors

The authors have successfully corrected the typographical errors previously mentioned. However, Figure 1 remains difficult to interpret due to its low resolution, as demonstrated by the attached zoomed-in image. To resolve this, the authors should either provide a higher-resolution version of the same figure or create a new set of images that convey the same information. As it stands, the current version of Figure 1 does not enhance the understanding beyond what is already provided in the descriptive text.

Author Response

I removed figure 1 and the reference to it in the text.

Reviewer 4 Report

Comments and Suggestions for Authors

I would suggest to emphasize the scientific merit and novelty of the approach described in the manuscript.

Author Response

I have added a comparison with the prior articles as well as the specific reason our report is different. This is in lines 181 - 185.